# Activation of Mast-Cell-Derived Chymase in the Lacrimal Glands of Patients with IgG4-Related Ophthalmic Disease

**DOI:** 10.3390/ijms23052556

**Published:** 2022-02-25

**Authors:** Yasushi Fujita, Denan Jin, Masashi Mimura, Yohei Sato, Shinji Takai, Teruyo Kida

**Affiliations:** 1Department of Ophthalmology, Osaka Medical and Pharmaceutical University, Takatsuki 569-8686, Japan; opt119@osaka-med.ac.jp (M.M.); satoyohei5795@yahoo.co.jp (Y.S.); teruyo.kida@ompu.ac.jp (T.K.); 2Department of Innovative Medicine, Osaka Medical and Pharmaceutical University, Takatsuki 569-8686, Japan; denan.jin@ompu.ac.jp (D.J.); shinji.takai@ompu.ac.jp (S.T.)

**Keywords:** chymase, mast cells (MCs), IgG4-related ophthalmic disease (IgG4-ROD), lacrimal glands (LGs), transforming growth factor-β1 (TGF-β1)

## Abstract

The purpose of this present study was to investigate the distribution and expression of chymase in the lacrimal glands (LGs) of patients afflicted with IgG4-related ophthalmic disease (IgG4-ROD). LGs from patients with severe canalicular obstruction were considered the control group. Toluidine blue staining confirmed a significant increase in the number of mast cells in the LGs obtained from the IgG4-ROD patients. In addition, immunostaining of serial sections from the LGs showed a significant increase in the number of chymase-positive cells and tryptase-positive cells in the IgG4-ROD LGs compared to the normal control LGs. The mRNA expression of chymase, tryptase, TGF-β1, and collagen-I tended to increase in the IgG4-ROD LGs. Immunostaining of vimentin and α-smooth muscle actin (α-SMA) showed that myofibroblasts were the main cellular components in severely fibrotic regions of LGs in patients with IgG4-ROD. Linear regression analyses on the number of mast cells, chymase-positive cells, and tryptase-positive cells revealed significant positive correlations between those respective cells. Our findings suggest that chymase may play a role in the fibrotic disorder of IgG4-ROD LGs through the regulation of TGF-β1 activation and collagen-I deposition, and that it may be a therapeutic target for patients afflicted with IgG4-ROD.

## 1. Introduction

IgG4-related disease (IgG4-RD) is a systemic disease of unknown etiology that causes infiltration of IgG4-positive plasma cells into various organs, thus resulting in severe fibrosis [1]. Previously, IgG4-RD was recognized as autoimmune pancreatitis in the field of gastroenterology [2]. However, IgG4-RD has recently become recognized as a relatively new disease concept, as, in addition to the typical target organs such as the pancreas, lacrimal glands (LGs), and salivary glands, it reportedly invades various other organs such as the hepatobiliary system, lungs, kidneys, prostate, and thyroid glands [3]. Recent Japanese studies have focused on specifying the criteria for the diagnosis of IgG4-RD [4,5,6].

Mikulicz disease, which is well known in the field of ophthalmology, is a disorder that is reportedly accompanied by swelling of the salivary glands and the LGs in both eyes [7]. Moreover, a previous study reported that Mikulicz disease causes swelling and fibrosis in other various orbital lesions, such as the extraocular muscles, the trigeminal nerve, and the optic nerve, and it has now come to be collectively recognized as an IgG4-related ophthalmic disease (IgG4-ROD) [8]. In patients afflicted with IgG4-ROD, pathological examination is necessary for diagnosis, and it is characterized by fibrosis forming a mass lesion mainly composed of an infiltration of IgG4-producing plasma cells and lymphocytes [9].

Although the exact pathophysiology of IgG4-RD has yet to be clarified, an autoimmune mechanism appears to be strongly related to this disorder [3]. For example, IgG4-RD is often associated with allergic conditions in many patients, and an increased number of IgE-positive mast cells has been observed in lymphoid, biliary, and pancreatic tissue samples of patients with IgG4-RD [10]. However, the exact precipitating factors for the pathogenesis of IgG4-RD have yet to be identified. Interestingly, it has been reported that, in cases of IgG4-RD, mast cells (MCs) are increased in the submandibular gland and that those MCs possibly produce some T helper 2 and regulatory T-cell cytokines, such as interleukin (IL)-4, IL-10, and IL-13 [11,12]. Traditionally, MCs can divide into MC_T_ (tryptase-positive, chymase-negative) and MC_TC_ (tryptase-positive, chymase-positive) types based on the differences in protease composition [13]; i.e., MC_T_ cells contain tryptase alone, while MC_TC_ cells contain both tryptase and chymase. Reportedly, both of these MC types release three groups of mediators upon degranulation, i.e., preformed mediators (e.g., histamine, tryptase, and chymase), de novo synthesized mediators (e.g., inflammatory-related leukotrienes (LTs) LTB4 and LTD4), and cytokines (e.g., IL-1β, tumor necrosis factor-alpha (TNF-α), and transforming growth factor beta-1 (TGF-β1)) [14]. Among those mediators, chymase has attracted much attention due to regulation of angiotensin (Ang) II generation, as well as activation of latent TGF-β1 to its active form [15,16]. On the other hand, chymase production of stem-cell factor (SCF) by enzymatic cleavage can yield a bioactive soluble product [17], thus suggesting that chymase may be very important in the regulation of MC numbers and maturation. Following the development of orally active chymase inhibitors, the inhibition of MC-derived chymase has been shown to be beneficial in the treatment of several organ fibrogenetic disorders in animal models [18]. Since IgG4-RD is often accompanied with fibrosis in various organs, and since, to the best of our knowledge, there have been no previous published reports on the relationship between chymase and IgG4-RD, the purpose of this present study was to compare the expression characteristics of chymase and its regulating fibrogenic mediators, as well as the quantities of chymase positive MCs, in LGs obtained from patients afflicted with IgG4-ROD and from patients diagnosed with severe canalicular obstruction yet normal LGs as a control.

## 2. Results

### 2.1. General Histological Examinations

Representative hematoxylin and eosin (H&E) staining and Azan Mallory staining images of the IgG4-ROD-patient LGs and the normal control LGs are shown in Figure 1 and Figure 2. As can be seen in the images, glandular structure was maintained in the control LGs; i.e., the trace of lachrymal fluid retentions were surrounded by the secretory epithelium cells (Figure 1, the bottom column of high magnification image) which gathered to form several lobes (Figure 1, the upper column of low magnification image). Conversely, the glandular structure in the LGs of the IgG4-ROD patients was completely destroyed by the chronic inflammation and was accompanied by marked focal lymphocytic hyperplasia (Figure 1, the upper column of low magnification image). In those lymphocytic hyperplasia areas, many plasma cells (black arrows) were scattered among infiltrated lymphocytes (Figure 1, the bottom column of high magnification image).

As can be seen in Figure 2, Azan Mallory staining revealed that the lobes of secretory epithelial cells in the control-subject LGs were surrounded by light-blue fibrotic interstitial tissue. However, the LGs in the IgG4-ROD patients showed a dark-blue staining in the surrounding tissue of focal lymphocytic hyperplasia areas, thus indicating the presence of severe fibrosis. Moreover, gene expressions of collagen-I were significantly elevated in the LGs from the patients with IgG4-ROD (Figure 2, bar graph).

Representative toluidine blue staining images of the IgG4-ROD-patient LGs and the normal control LGs, as well as the quantitation of MC numbers between the two groups, are shown in Figure 3. As can be seen in the images, there were a few toluidine blue-positive MCs in the control LG (Figure 3, black arrows). However, a marked accumulation of MCs was observed in the LGs of the IgG4-ROD patients (Figure 3, black arrows), and there was a statistically significant increase in the number of MCs in the IgG4-ROD patients compared with that in the control subjects (Figure 3, bar graph).

### 2.2. Identification of Mast Cell Type by Immunohistoolgical Examinations

Representative tryptase immunostaining images of the IgG4-ROD-patient LGs and the normal control LGs, as well as the quantitation of tryptase-positive cell numbers and the mRNA expression of tryptase between the two groups, is shown in Figure 4. As with the expression of MCs in the control LG, the accumulation of tryptase-positive cells was small (Figure 4, black arrows). However, there was a large accumulation of those cells in the LGs of the IgG4-ROD patients (Figure 4, black arrows). In comparison with the control LGs, the gene expression of tryptase and the numbers of tryptase-positive cells were markedly increased in the LGs of the IgG4-ROD patients (bar graphs in Figure 4). As indicated in the line graph of Figure 3, a strong positive correlation was found between the numbers of MCs and tryptase-positive cells. Similar results were also observed in the chymase-related examinations.

As can be seen in Figure 5, only a few chymase-positive cells were observed in the control LGs (Figure 5, black arrows). However, a marked accumulation of chymase-positive cells was observed in the LGs of the IgG4-ROD patients (Figure 5, black arrows), and that increase was statistically significant in comparison to the increase in the control LGs (Figure 5, bar graph). Moreover, the gene expression of chymase tended to increase in the LGs of the IgG4-ROD patients. It should be noted that in the present study, linear regression analysis was also performed to investigate the relationships among the numbers of MCs, tryptase-positive cells, and chymase-positive cells. As indicated in the line graph in Figure 4, a significant positive correlation was observed between the number of tryptase-positive cells and chymase-positive cells. Correlation analysis also showed a significant positive correlation between MCs and chymase-positive cells (line graph in Figure 5), indicating that the majority of mast cells contain both tryptase and chymase in the LGs obtained from both control and the IgG4-ROD patients.

### 2.3. Distribution Pattern of TGF-β1 in the LGs of the IgG4-ROD

Representative TGF-β1 immunostaining images of the control LGs and representative TGF-β1 and chymase immunostaining, as well as Azan Mallory staining performed on the serial sections of the LGs of the IgG4-ROD patients, are shown in Figure 6. The number of TGF-β1-positive cells in the control LGs was small (Figure 6, black arrows). However, the number of those cells was largely increased in the LGs of the IgG4-ROD patients (Figure 6, black arrows). Moreover, TGF-β1-positive staining was mainly observed in fibroblast-like cells, and in some cases, MCs also showed positive staining for TGF-β1. As can be seen in Figure 6, the spots of chymase-positive staining were overlapped with the spots of TGF-β1-positive staining, and as mentioned above, these staining patterns were observed in the serial sections, thus indicating that the MC_TC_ type of MCs was also the cellular source for TGF-β1. The gene expression of TGF-β1 tended to increase in the LGs of the IgG4-ROD patients. As shown in the results of the Azan Mallory staining performed in the serial section neighboring the TGF-β1 staining, TGF-β1- and chymase-positive cells were expressed mainly in the fibrotic areas (Figure 6).

### 2.4. Identification of Fibroblasts and Myofibroblasts in the Fibrotic Regions of LGs

Representative vimentin and α-SMA immunostaining performed on serial sections of the LGs obtained from patients with canalicular obstruction or IgG4-ROD are shown in Figure 7. These loci were matched to the yellow frames of Azan Mallory-stained sections shown in the left column. As mentioned in Figure 2, light-blue staining was observed in the lobes of secretory epithelial cells surrounding interstitial tissue from control LGs, indicating the presence of relatively low collagen deposition in these regions. On the other hand, in the LGs of IgG4-ROD patients, dark-blue staining was found in the tissues surrounding areas of focal lymphocytic hyperplasia, indicating the presence of a large amount of collagen deposition, as well as severe fibrosis. In the present study, we also performed immunostaining in the LGs using antibodies against human vimentin and α-SMA to investigate whether the phenomenon of TGF-β1 dependent fibroblast activation can be found in severely fibrotic regions of the LGs of IgG4-ROD patients. As is well known, vimentin is a marker protein for cells of mesenchymal origin that is mainly expressed in fibroblasts and myofibroblasts, as well as endothelial cells [19]. While α-SMA is a contractile protein that is mainly expressed in contractile vascular smooth muscle cells, it is also expressed in myofibroblasts after the phenotypic change from fibroblasts has occurred [20]. Therefore, the proportion of fibroblasts and myofibroblasts in the interstitial tissue of LGs can be calculated by the combination of vimentin and α-SMA immunostaining. On the other hand, the number of fibroblasts and ratio of fibroblasts to myofibroblasts in the interstitial tissue of LGs might partially reflect the degree of latent TGF-β1 activation by chymase. It is well known that activated TGF-β1 not only stimulates fibroblast proliferation and collagen production, but also accelerates the differentiation of fibroblasts into myofibroblasts [21]. In the present study, vimentin-positive staining was diffusely scattered in the interstitial tissue of control LGs, while no α-SMA-positive staining could be found in the serial sections next to the vimentin staining, indicating that all these vimentin-positive cells are to be the fibroblasts (upper images of Figure 7). In contrast, in the LGs of IgG4-ROD patients, vimentin-positive cells were densely distributed in the surrounding fibrotic areas of focal lymphocytic hyperplasia. Further, α-SMA immunostaining performed in serial sections adjacent to the vimentin immunostained areas confirmed that most of these α-SMA-positive cells overlapped with the vimentin-positive cells, indicating that most of these vimentin-positive cells were myofibroblasts. Since chymase and TGF-β1 positive staining was also frequently observed in the severe fibrotic areas (Figure 6), the increase in myofibroblasts observed in the areas surrounding the fibrotic areas of focal lymphocytic hyperplasia in the present study might have been a result of chymase-dependent TGF-β1 activation.

## 3. Discussion

IgG4-ROD is generally an indolent chronic process, and it can progress slowly over many years and evolve from a single organ to multiple organ involvement. IgG4-ROD can present in many ways, yet the most common patterns include bilateral LG, trigeminal lesions, extraocular muscle swelling, and sclerosing orbital inflammation [22]. If proper treatment is not applied or if treatment is delayed, the disease process can lead to organ damage and loss of function. The standard treatment for IgG4-ROD is systemic corticosteroids [3], as corticosteroids elicit an excellent response during primary therapy. However, the therapeutic effect of corticosteroids is unsustainable, and long-term administration is necessary to prevent disease recurrence. As is well known, the long-term administration of corticosteroids can cause various complications, such as diabetes, weight gain, osteoporosis, and gastric ulcer [23,24]. Therefore, the development of new pharmacological therapies for such disorders is urgently needed.

In this present study, our findings revealed a significant increase in the number of MCs, chymase-positive cells, and tryptase-positive cells in the LGs obtained from patients afflicted IgG4-ROD in comparison with the normal LGs obtained from control subjects. Moreover, gene expression of chymase and tryptase in the LGs of patients afflicted IgG4-ROD was also markedly increased. In view of the findings that the chymase-positive cell numbers were positively correlated with the tryptase-positive cell numbers and that there were no significant differences between the chymase- and tryptase-positive cell numbers in the LGs of the patients afflicted with IgG4-ROD (data not shown), we theorize that that the increase in MCs in the LGs obtained from patients afflicted with IgG4-ROD mainly results from the increase in the MC_TC_ type of MCs. As we mentioned above, and in accordance with whether or not the MCs contain chymase or tryptase, these cells can divide into MC_T_ and MC_TC_ types, and as has been reported, the MC_TC_ type contains both chymase and tryptase [13].

Recently, MC-derived chymase has attracted great attention due to its Ang II- and TGF-β1-generating properties. For example, chymase can powerfully cleavage Ang I to Ang II [15], and, as is quite widely well known, Ang II is not only a powerful vasoconstrictor but also a key player in some inflammatory processes [25]. For example, Ang II can increase vascular permeability through the release of prostaglandins and vascular endothelial cell growth factor and can contribute to the recruitment of inflammatory cells into the tissue via the regulation of adhesion molecules and chemokines by resident cells. Moreover, Ang II appears to be important in the fibrogenic pathology. It was previously reported that Ang II can interact with TGF-β1 to promote collagen deposition [26], which is a very important process in organ fibrosis. On the other hand, chymase activates latent TGF-β1 to produce mature TGF-β1 [16] and TGF-β1 increases collagen production and stimulates fibroblasts to differentiate to the myofibroblasts [27].

In this present study, our findings revealed that chymase- and TGF-β1-positive cells were expressed mainly along the collagen-rich fibrotic region (Figure 6), and that the spots of chymase-positive staining were overlapped partially with the spots of TGF-β1-positive staining, thus indicating that chymase is able to cleavage the latent TGF-β1 to its active form upon these positions. Thus, the increase in the numbers of chymase-positive MCs denotes the activation of chymase, which in turn increases the quantities of mature TGF-β1, thus resulting in fibroblasts possibly being phenotypically changed to myofibroblasts. It is well known that the capacity of myofibroblasts to produce collagen is several times greater than fibroblasts. Thus, the inhibition of chymase via a pharmacological method can not only reduce Ang II generation but also decrease active TGF-β1 production, and all of these responses may be beneficial in the reduction of inflammatory and fibrogenic response in the LGs of patients afflicted with IgG4-ROD. It should be noted that those beneficial effects do not apply to IgG4-ROD alone, as Takai et al. previously reported that chymase inhibition through the attenuation of Ang II, TGF-β1, collagen I, and matrix metalloproteinase 9 production improved several fibrogenic and inflammatory disorders in animal models [18]. For example, the inflammatory and fibrogenic pathophysiological conditions were largely suppressed with the treatment of chymase-specific inhibitor TY-51469 in animal-model experiments, resembling nonalcoholic steatohepatitis [28], inflammatory bowel disease [29], and intraperitoneal adhesions [30]. Unfortunately, we could not examine the beneficial effects of the chymase inhibitor in an animal model resembling IgG4-ROD, as such a model does not currently exist. Thus, the pathophysiological roles of chymase in IgG4-ROD require further investigation involving the use of chymase inhibitors in novel IgG4-ROD animal models.

## 4. Limitations

Since surgical resection of LGs for the treatment of both severe canalicular obstruction and IgG4-ROD is rarely performed at our hospital, we could not enroll a satisfactory number of subjects in the present study. Additionally, the present study focused only on the features of the histological distribution and the degree of chymase expression between the two groups. However, since mast cells contain several mediators, possible differences in these other mediators between the groups might also have affected the study results. Therefore, one cannot reach a final conclusion until beneficial effects are identified using a chymase-specific inhibitor in animal models simulating the pathophysiology of IgG4-ROD.

## 5. Materials and Methods

### 5.1. Collection of LGs

A total of 12 LG samples were collected for this present study, i.e., 7 LGs obtained via biopsy from patients who were definitively diagnosed as IgG4-ROD according to the pathological characteristics (2 males and 5 females, mean age: 52.4 ± 10.7 years, range: 39–72 years) and 5 LGs to be used as normal control samples that were obtained from non-IgG4-ROD patients diagnosed with severe canalicular obstruction except for tumor, trauma, and drug influence such as tegafur (3 males and 2 females, mean age: 76.6 ± 5.8 years, range: 67–85 years). Clinically, conjunctivodacryocystorhinostomy using a Jones tube is the technique that is usually selected for the treatment of severe canalicular obstruction. However, there are some complications that can occur via this method, such as foreign body sensation, prolapse, invasion, and infection by Jones tube [31]. Thus, we resect partial LGs for the improvement of epiphora in these patients as a last resort. These samples were grouped as the control in this study.

The protocols of this study were approved by the Research Ethics Committee (Approval No. 2020-067) and the Human Studies Committee of Osaka Medical and Pharmaceutical University, Takatsuki-City, Osaka, Japan, and in accordance with the tenets set forth in the Declaration of Helsinki, informed consent was obtained from all patients prior to their involvement in the study.

### 5.2. General Histological and Immunohistological Studies

Paraffin blocks were prepared from the formalin-fixed LGs, and 3-μm serial cross-sections were prepared for histological staining. The first serial cross-sections from each of the paraffin blocks were stained with H&E to identify the infiltrated lymphocytes. Azan Mallory staining was performed on the second cross-sections to identify fibrosis. H&E and Azan Mallory staining were performed in accordance with the standard staining protocols. The fourth cross-sections were stained with toluidine blue to identify MC distribution. In brief, deparaffinized sections were immersed in 0.5% toluidine blue solution (pH 4.8) for approximately 15 min, fractionated with 0.5% glacial acetic acid solution, and then mounted after drying.

The third, fourth, and fifth serial cross-sections were used to reveal the distribution of tryptase and chymase by using anti-tryptase antibody (M7052, 1:800 dilution; Dako Denmark A/S, Glostrup, Denmark) and anti-chymase antibody (mouse monoclonal antibody against human MC chymase, 2D11G10D, 1:1000 dilution; a kind gift from Dr. Takeo Suzuki, Katakura Industries Co., Ltd., Saitama, Japan), respectively. To evaluate the mesenchymal cellular components (such as fibroblasts and myofibroblasts) among LGs’ tissues, vimentin (1:100 dilution; Cell Signaling Technology, Danvers, MA, USA) and α-smooth muscle actin (α-SMA) (1:200 dilution; Dako) immunostainings were performed on the sixth and seventh sections. Immunohistological staining using the above-mentioned antibodies was performed in accordance with protocols described elsewhere [32]. In brief, deparaffinized sections were incubated with respective antibodies overnight at 4 °C, followed by reaction with components from a labeled streptavidin–biotin peroxidase kit (LSAB^®^ System-HRP; Dako North America, Inc., Carpinteria, CA, USA) that included 3-amino-9-ethylcarbazole for color development. Finally, the count staining for these sections was performed with hematoxylin, with the sections then mounted with cover glasses.

To analyze the cell numbers in the two groups, the cellular number in each cross section was counted at HPF (200×) in the three densest areas (i.e., the hot spots), with the mean value then being used for statistical analysis.

### 5.3. Real-Time Polymerase Chain Reaction (RT-PCR)

To extract the RNA of the LGs, 10 sheets of the 20-μm-thickness paraffin sections were collected from the respective formalin-fixed paraffin-embedded tissue blocks using a microtome (LITORATOMU, REM-710; Yamato Koki Kogyo Co., Ltd., Himeji, Japan). Total RNA was extracted in accordance with the protocol provided in the total RNA isolation kit (ISOGEN PB Kit; Nippon Gene Co., Ltd., Tokyo, Japan).

Total RNA (1 μg) was transcribed into cDNA via the use of a SuperScript™ VILO™ cDNA Synthesis Kit (Invitrogen™ Corporation, Carlsbad, CA, USA). Then, mRNA levels were measured by RT-PCR on a Stratagene Mx3000P qPCR System (Agilent Technologies, Inc., Santa Clara, CA, USA) using Taq-Man™ (Applied Biosystems^®^, Waltham, MA, USA) fluorogenic probes. RT-PCR primers and probes for TGF-β1, collagen-I, tryptase, chymase, and glyceraldehyde-3- phosphate dehydrogenase (GAPDH) were designed by Roche Diagnostics K.K. (Tokyo, Japan). The primers were 5′-cattaacgggttcagttccag-3′ (forward) and 5′-agcaggaagggtcggttc-3′ (reverse) for TGF-β1, 5′-caacagccgcttcacctac-3′(forward) and 5′-caggctccggtgtgactc-3′(reverse) for collagen-I, 5′-gatgctgagcctgctgct-3′ (forward) and 5′-gacgatacccgcttgctg-3′ (reverse) for tryptase, 5′-atccctcagacccaagagg-3′ (forward) and 5′-ggaagctggatctttattgagg-3′ (reverse) for chymase, and 5′-aatgtatcagttgtggatctgacc-3′ (forward) and 5′-gcttcactaccttcttgatgtcg-3′ (reverse) for GAPDH. The mRNA levels of TGF-β1, chymase, collagen-I, and tryptase were normalized to those of GAPDH.

### 5.4. Statistical Analysis

All numerical data are expressed as the mean ± SEM. Significant differences between the mean values of the groups were evaluated via the Mann–Whitney U test. Pearson’s correlation coefficient was measured to test the linear relationship between two variables using linear regression analysis. A *p*-value of <0.05 was considered statistically significant.

## 6. Conclusions

The results of the present study revealed a marked increase in chymase-positive cells, as well as chymase gene expression, in LGs obtained from patients with IgG4-ROD in comparison with those in normal control LGs. Furthermore, the distribution pattern of chymase in the LG lesions was extremely similar to that of TGF-β1, and the location of the increase in myofibroblasts observed in the surrounding fibrotic areas of focal lymphocytic hyperplasia in the present study was very similar to the distribution of chymase and TGF-β1, indicating that chymase might play a role in the fibrotic disorder of LGs in patients with IgG4-ROD through regulation of TGF-β1 activation and collagen-I deposition.

IgG4-RD is a systemic disease and usually involves fibrosis in various organs. Currently, apart from steroid treatment, no other effective pharmacological treatments are available to treat this condition. Therefore, development of clinically useful chymase specific inhibitors might be a new therapeutic strategy for the treatment of IgG4-RD, including IgG4-ROD.

## Figures and Tables

**Figure 1 ijms-23-02556-f001:**
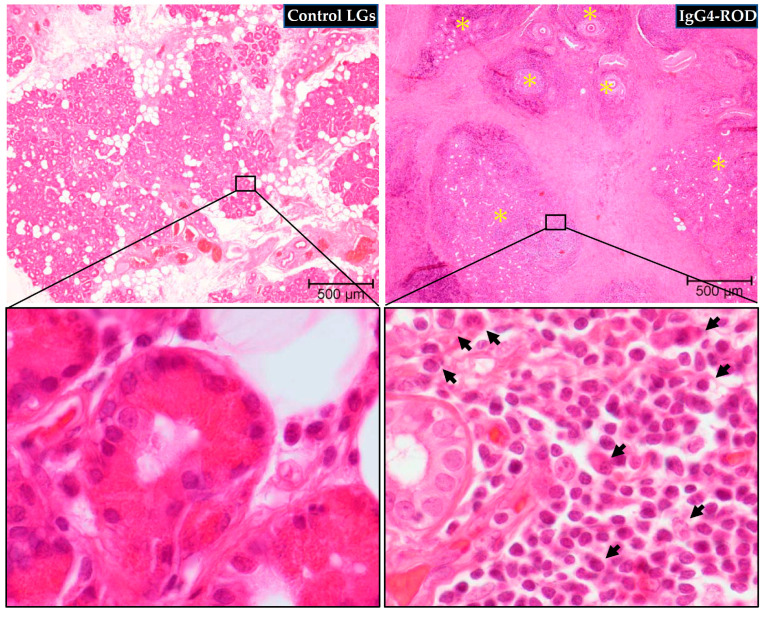
Representative images of hematoxylin and eosin (H&E) staining of the LGs obtained from patients with IgG4-ROD and patients diagnosed with severe canalicular obstruction yet normal LGs as a control. Glandular structure was maintained in the normal control LGs, while completely destroyed in the LGs of the IgG4-ROD patients. Yellow asterisks indicate the area of lymphocytic hyperplasia. Black arrows indicate plasma cells.

**Figure 2 ijms-23-02556-f002:**
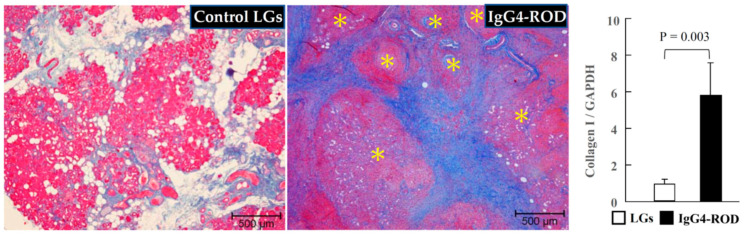
Representative images of Azan Mallory staining and collagen-I expressions of the LGs obtained from the patients with IgG4-ROD and from the patients diagnosed with severe canalicular obstruction yet with no IgG4-ROD. In the LGs of the normal control subjects, the lobes were surrounded by light-blue fibrotic interstitial tissue, while a dark-blue staining in the lymphocytic hyperplasia focus surrounding tissues was observed in the LGs from the patients with IgG4-ROD. Yellow asterisks indicate the area of lymphocytic hyperplasia. Bar graphs show the gene expressions of collagen-I in the LGs of the two groups.

**Figure 3 ijms-23-02556-f003:**
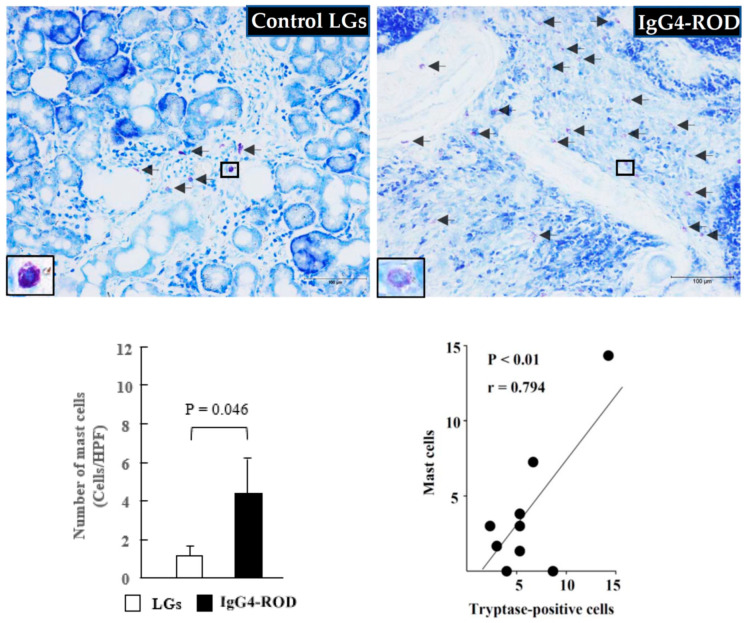
Representative images of toluidine blue staining and MCs population of the LGs obtained from the patients with IgG4-ROD and from the control patients diagnosed with severe canalicular obstruction yet with no IgG4-ROD. Black arrows indicate MCs. Bar graph shows the quantitation of the MCs numbers between the two groups. Line graph shows the linear regression analysis between the numbers of MCs and tryptase-positive cells in all data of the examined LGs.

**Figure 4 ijms-23-02556-f004:**
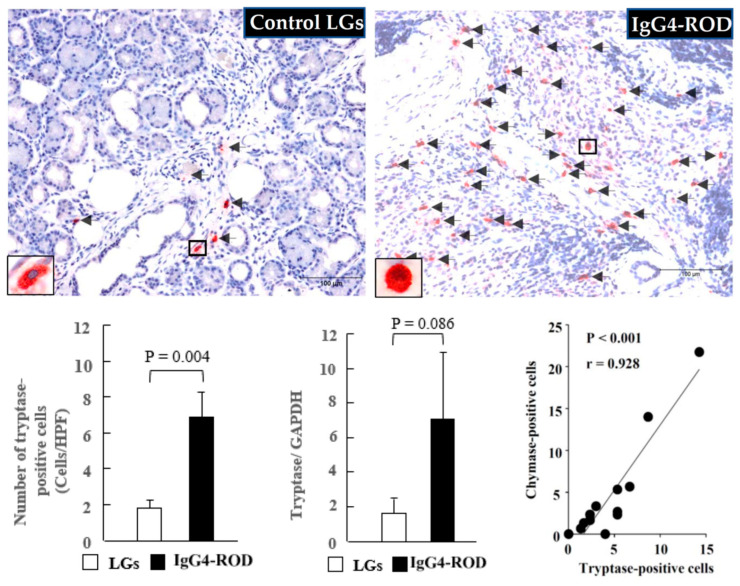
Representative images of tryptase immunostaining, the tryptase-positive cell population, and the tryptase expression of the LGs obtained from the patients with IgG4-ROD and from the control patients diagnosed with severe canalicular obstruction yet with no IgG4-ROD. Bar graphs show the quantitation of the tryptase-positive cell numbers and the tryptase expression between the two groups. Line graph shows the linear regression analysis between the numbers of tryptase-positive cells and chymase-positive cells in all data of the examined LGs. Black arrows indicate the tryptase-positive cells.

**Figure 5 ijms-23-02556-f005:**
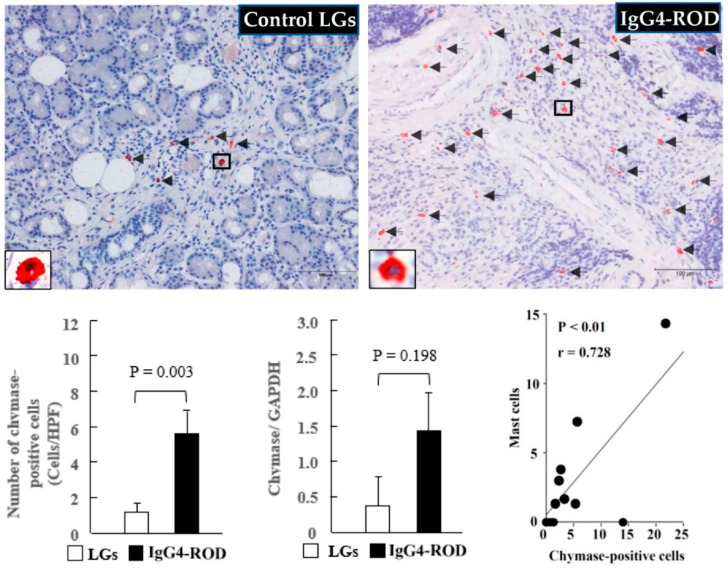
Representative images of chymase immunostaining, the chymase-positive cell population, and the chymase expression of the LGs obtained from the patients with IgG4-ROD and from the control patients diagnosed with severe canalicular obstruction yet with no IgG4-ROD. Black arrows indicate chymase-positive cells. Bar graphs show the quantitation of the chymase-positive cell numbers and the chymase expression between the two groups. Line graph shows the linear regression analysis between the numbers of chymase-positive cells and MCs in all data of the examined LGs.

**Figure 6 ijms-23-02556-f006:**
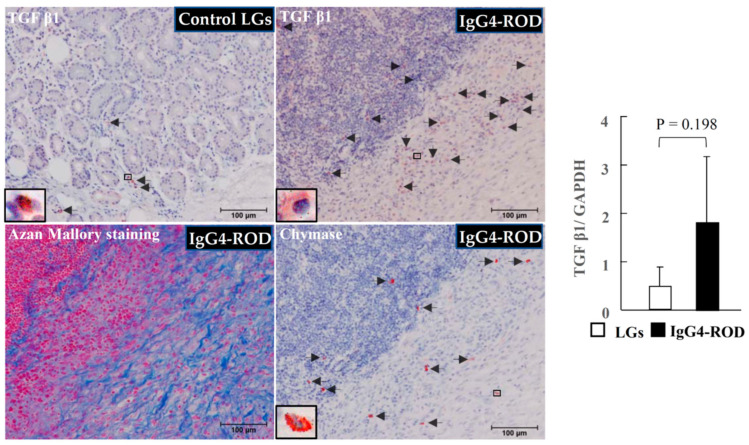
Representative images of transforming growth factor beta-1 (TGF-β1) immunostaining of the normal control LGs, and representative images of TGF-β1 and chymase immunostaining, as well as Azan Mallory staining, performed on serial sections of the LGs obtained from the IgG4-ROD patients. Black arrows indicate the TGF-β1-positive cells or the chymase-positive cells. Bar graphs show the gene expression of TGF-β1 in the LGs in two groups.

**Figure 7 ijms-23-02556-f007:**
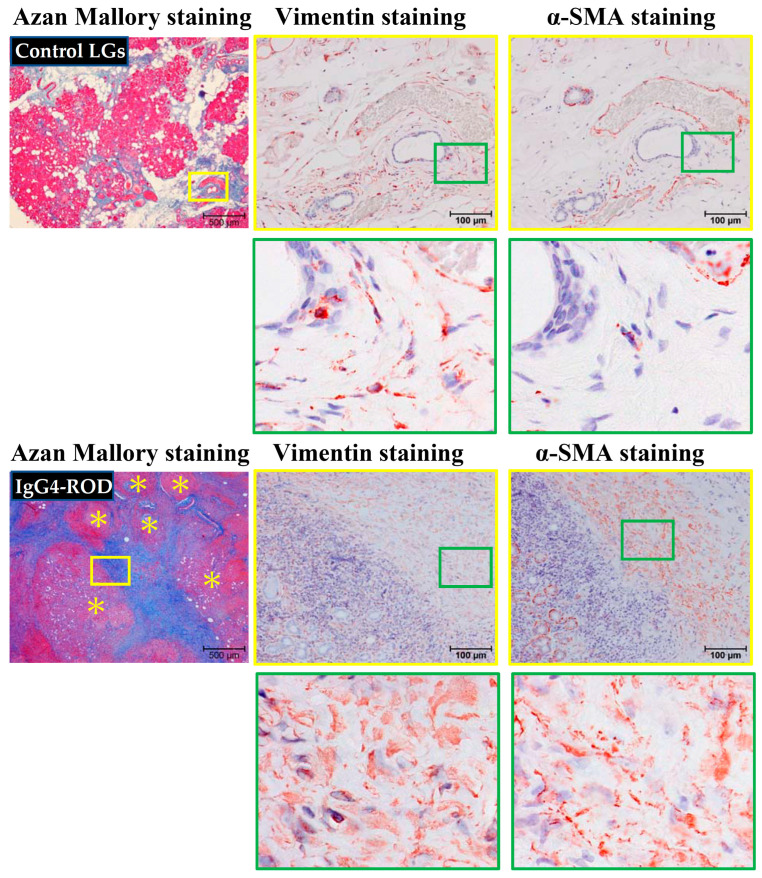
Representative images of vimentin and α-SMA immunostaining performed on serial sections of LGs obtained from patients with IgG4-ROD and from control patients diagnosed with severe canalicular obstruction, but with no IgG4-ROD. These loci were matched with the yellow frames of Azan Mallory-stained sections, shown in the left column. The images of green frames in the bottoms are indicating the enlargement of loci surrounded by green squares. Yellow asterisks indicate the area of lymphocytic hyperplasia.

## Data Availability

The data that support the findings of this study are available on request from the corresponding author, Yasushi Fujita (E-mail: yasushi.fujita@ompu.ac.jp).

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
