# Peer review of "Activation of Mast-Cell-Derived Chymase in the Lacrimal Glands of Patients with IgG4-Related Ophthalmic Disease"

_ijms, 2022, doi:10.3390/ijms23052556_

Round 1

Reviewer 1 Report

I would like to congratulate the authors the manuscript as the material used and the topic in general is very interesting and could be applied also to other conditions associated with local inflammation involving mast cells and fibroblasts. However, I would have several comments and suggestions that could significantly improve quality of the paper and make it more suitable for publication in the International Journal of Molecular Sciences.

  1. Considering the fact that apart form TGF-beta and chymase also tryptase was analyzed, at least short association of that parameter with studied pathological condition should be described.
  2. The authors might perform in-depth screening of the introduction and discussion section, especially to make sure that there are no any spelling errors, unnecessary repetitions, and the sentences syntax that could affect manuscript quality.
  3. I would consider Abstract well written, however, the limit for that part is 200 words so I would kindly suggest to make it more concise through reduction to the limit required by the journal.
  4. Would it be possible for the authors to provide more comprehensive, tabular for example, presentation of the study groups? In addition, data from diagnostic procedures would be beneficial for the paper – including lab tests results and physical examination. In accordance, data from clinical diagnostic procedures could be used for their correlation with the results of the research analyses described in the manuscript.
  5. Quality of figure 1 could be improved and, in addition, there is no need of indicating ‘low/high magnification if there would be exact magnification value in the figure legend. Scale for low magnification photos should be made more readable.
  6. Tabular presentation of pathological analysis of the samples would be beneficial for the study groups presentation as clear justification of the conditions and their usefulness for use in the presented study.
  7. In my opinion simple analysis of associations, for example between number of mast cells and selected marker, could be provided on graphs together with microscopic and gene expression analysis data. In that way, on one graph immunohistochemistry, PCR and correlation results for selected parameter would be clearly presented on one graph.
  8. In reference to gene expression analysis, exact indication of values presented (delta Ct or 2^(-deltaCt) could be included in the brackets apart from marker/control on y-axis title. In addition to gene expression results, I think that correlation of these parameters with microscopic and clinical data should also be presented.
  9. As additional comment regarding correlation analysis, for larger number of parameters (including clinical data) I would recommend presentation of theses analyses as heat-maps, with r value of each comparison and indication of the statistical significance for example with asterisks in each bracket of the map.
  10. Despite photos from immunohistochemical staining in figure 7, there is only graphical presentation of the TGF-beta gene expression analysis. The figure should also be supplemented with results from the tissue samples microscopic analysis.
  11. Due to my suggestions for different organization of the data in graphs, the legends should also be rewritten to concisely provide full description of the figures. Use of sections within figures (indicated for example with letters) would be helpful to prepare clear, readable figure legends.
  12. Could the authors indicate within the Statistical Analysis subsection why certain statistical tests were used? That would be beneficial to justify whether the right biostatistical approach was used. In my opinion it seems that when using non-parametric tests, the data should be presented as medians and interquartile ranges. Moreover, in case of correlation analysis Spearman test might be more proper considering low number of samples and possibly lack of Gaussian distribution.
  13. Considering suggestion that mast cells-derived chymase might increase TGF-beta1 levels locally, thus, possibly inducing change of fibroblasts phenotype into myofibroblasts, did authors think of analysis of myofibroblasts distribution/density in the tissue areas of chymase increased activity?
  14. Despite tryptase analysis was also included within the study, that aspect is not comprehensively discussed and referred to current state of knowledge and possible applications of the data.
  15. I think that limitations of the study should be stated by the authors within Discussion section. Moreover, more highlighted ending summary could be beneficial at the end of that section.
  16. The authors could have a closer look at the Conclusion section to make it more readable through using less complex sentences and providing one decisive opinion on the study significance. Furthermore, completed by possible implementation of the data and future directions.

Author Response

Feb 17, 2022

Editorial Office

Dear Dr. Reviewer 1:

Thank you for your detailed review of our manuscript. We have revised our manuscript in accordance with your suggestions and recommendations, and have provided point by point responses to your comments below. We believe that the corrections will make our paper much more valuable and mature. Our responses to your comments are as follows:

  • As you rightly stated, besides TGF-beta and chymase, mast cells also contain several other mediators, including tryptase. Therefore, we have added a limitation paragraph in the revised manuscript to avoid misleading the readers as follows (Lines 291-299):

Since surgical resection of LGs for the treatment of both severe canalicular obstruction and IgG4-ROD is rarely performed at our hospital, we could not enroll a satisfactory number of subjects in the present study. Additionally, the present study focused only on the features of the histological distribution and the degree of chymase expression between the two groups. However, since mast cells contain several mediators, possible differences in these other mediators between the groups might also have affected the study results. Therefore, one cannot reach a final conclusion until beneficial effects are identified using a chymase-specific inhibitor in animal models simulating the pathophysiology of IgG4-ROD.

  • We have carefully re-read our paper and deleted unnecessary clinical interpretations about IgG4-ROD in the Discussion section. Additionally, we have modified our previous discussion to some extent.
  • We have shortened the Abstract to below 200 words.
  • We are sorry, but we could not find appropriate information to satisfy the reviewer’s recommendations in this comment.
  • As you mentioned, the quality of the images in Figure 1 was indeed poor. We have retaken the pictures and have revised Figure 1 in this version of the manuscript.
  • We are sorry, but we could not find appropriate information to satisfy the reviewer’s recommendations in this comment.
  • We agree with your suggestion that the addition of correlation line graph to the respective relating Figures can clearly present the results. Therefore, we have deleted the previous correlation line graphs and incorporated the information in Figures 3, 4 and 5 of the present manuscript.
  • We are sorry, but we could not find appropriate information to satisfy the reviewer’s recommendations in this comment.
  • We are sorry, but we could not find appropriate information to satisfy the reviewer’s recommendations in this comment.
  • Unfortunately, we could not perform a quantitative analysis of TGF-beta1 in the immunohistochemical study. As seen in Figure 6, unlike chymase and tryptase immunostaining, TGF-beta1 is expressed in several types of cells, such as fibroblast-like cells and mast cells, and the color of the positive staining was not very Therefore, it was difficult to precisely quantitatively analyze TGF-beta1 expression by color extraction software.
  • As recommended, we have rewritten the figure legends according to the changes in the figures.
  • As far as we known, the Mann-Whitney U test can be used to assess the presence of a statistically significant difference between two groups, and it can be used regardless of whether the samples are normally or non-normally distributed. Therefore, the Mann-Whitney U test was used in the present study. On the other hand, since we only have software for the Pearson test, we analyzed the results of the present study using this test.
  • Thank you very much for your valuable suggestion. As you rightly indicated, it is important to provide evidence that chymase-dependent TGF beta 1 is involved in the pathogenesis of LGs fibrosis in IgG4-ROD Although a definitive conclusion can only be reached by evaluating the beneficial effects of using a chymase-specific inhibitor in animal models simulating the pathophysiology of IgG4-ROD and with clinical studies, identification of fibrosis-related processes in the region close to the area of distribution of increased chymase and TGF beta 1 would help support our present opinion. Therefore, we performed an additional experiment to determine if certain fibrosis related-processes, such as phenotypic changes from fibroblasts to myofibroblasts, could be found in the areas of activated chymase and TGF beta 1. These results have been added to the present revised manuscript as follows (Lines 185-218):

2.4 Identification of fibroblasts and myofibroblasts in the fibrotic regions of LGs

Representative vimentin and α-SMA immunostaining performed on serial sections of the LGs obtained from patients with canalicular obstruction or IgG4-ROD are shown in Figure 7. These loci were matched to the yellow frames of Azan–Mallory-stained sections shown in the left column. As mentioned in Figure 2, light blue staining was observed in the lobes of secretory epithelial cells surrounding interstitial tissue from control LGs, indicating the presence of relatively low collagen deposition in these regions. On the other hand, in the LGs of IgG4-ROD patients, dark-blue staining was found in the tissues surrounding areas of focal lymphocytic hyperplasia, indicating the presence of a large amount of collagen deposition, as well as severe fibrosis. In the present study, we also performed immunostaining in the LGs using antibodies against human vimentin and α-SMA, to investigate if the phenomenon of TGF-β1 dependent fibroblast activation can be found in severely fibrotic regions of the LGs of IgG4-ROD patients. As is well known, vimentin is a marker protein for cells of mesenchymal origin that is mainly expressed in fibroblasts and myofibroblasts, as well as endothelial cells [a18]. While α‐SMA is a contractile protein that is mainly expressed in contractile vascular smooth muscle cells, it is also expressed in myofibroblasts after the phenotypic change from fibroblasts has occurred [a19]. Therefore, the proportion of fibroblasts and myofibroblasts in the interstitial tissue of LGs can be calculated by the combination of vimentin and α‐SMA immunostaining. On the other hand, the number of fibroblasts and ratio of fibroblasts to myofibroblasts in the interstitial tissue of LGs might partially reflect the degree of latent TGF-β1 activation by chymase. It is well known that activated TGF-β1 not only stimulates fibroblast proliferation and collagen production, but also accelerates the differentiation of fibroblasts into myofibroblasts (a20). In the present study, vimentin-positive staining was diffusely scattered in the interstitial tissue of control LGs, while no α‐SMA-positive staining could be found in the serial sections next to the vimentin staining, indicating that all these vimentin-positive cells are to be the fibroblasts (upper images of Figure 7). In contrast, in the LGs of IgG4-ROD patients, vimentin-positive cells were densely distributed in the surrounding fibrotic areas of focal lymphocytic hyperplasia. Further, α‐SMA immunostaining performed in serial sections adjacent to the vimentin immunostained areas confirmed that most of these α‐SMA-positive cells overlapped with the vimentin-positive cells, indicating that most of these vimentin-positive cells were myofibroblasts. Since chymase and TGF-β1 positive staining was also frequently observed in the severe fibrotic areas (Figure 6), the increase of myofibroblasts observed in the areas surrounding the fibrotic areas of focal lymphocytic hyperplasia in the present study might have been a result of chymase-dependent TGF-β1 activation.

  • As you mentioned, apart from TGF-beta and chymase, mast cells also contain several other mediators, including tryptase. However, since we did not assess this in detail, we have added this as a limitation in this revised manuscript. Please also see our response to comment 1 regarding this.

(15) and (16)  As recommended, we have rewritten the conclusion as follows (Lines :373-385) :

The results of the present study revealed a marked increase in chymase-positive cells, as well as chymase gene expression, in LGs obtained from patients with IgG4-ROD in comparison with those in normal control LGs. Furthermore, the distribution pattern of chymase in the LG lesions was extremely similar to that of TGF-β1, and the location of the increase in myofibroblasts observed in the surrounding fibrotic areas of focal lymphocytic hyperplasia in the present study was very similar to the distribution of chymase and TGF-β1, indicating that chymase might play a role in the fibrotic disorder of LGs in patients with IgG4-ROD through regulation of TGF-β1 activation and collagen-I deposition.

IgG4-RD is a systemic disease and usually involves fibrosis in various organs. Currently, apart from steroid treatment, no other effective pharmacological treatments are available to treat this condition. Therefore, development of clinically useful chymase specific inhibitors might be a new therapeutic strategy for the treatment of IgG4-RD, including IgG4-ROD.

Thank you very much for your kind consideration.

We look forward to the appearance of our revised manuscript in International Journal of Molecular Sciences.

Sincerely yours,

Yasushi Fujita, M.D.

Department of Ophthalmology, Osaka Medical and Pharmaceutical University, 2-7 Daigaku-machi, Takatsuki, Osaka 569-8686, Japan.

                     TEL: +81-72-683-1221 (Ext 8475)

                     FAX: +81-72-684-6730

                     E-mail: yasushi.fujita@ompu.ac.jp

Reviewer 2 Report

Actually, nothing wrong will happen if this manuscript us published as is.

Author Response

Feb 17, 2022

Editorial Office

Dear Dr. Reviewer 2:

Thank you very much for your E-mail of Feb 02 regarding our manuscript “Activation of Mast-Cell-Derived Chymase in the Lacrimal Glands of Patients with IgG4-Related Ophthalmic Disease” (Manuscript ID: ijms-1579696).

We are deeply grateful to you for the positive evaluation of our paper and recommendation for publication in the IJMS.

Thank you very much for your kind consideration.

We look forward to the appearance of our revised manuscript in International Journal of Molecular Sciences.

Sincerely yours,

Yasushi Fujita, M.D.

Department of Ophthalmology, Osaka Medical and Pharmaceutical University, 2-7 Daigaku-machi, Takatsuki, Osaka 569-8686, Japan.

                     TEL: +81-72-683-1221 (Ext 8475)

                     FAX: +81-72-684-6730

                     E-mail: yasushi.fujita@ompu.ac.jp

Reviewer 3 Report

This study by Fujita et al examines a potential role of mast cells and chymase in the fibrotic process of the IgG4-related ophthalmic disease.In this context, chymase could be a promising drug target for the development of effective antifibrotic therapies. The article is easy to understand and interesting. Such paper is welcome to highlight the need for further studies on this field. Yet, I would like to point out the following comments.

Major comments
1.The number of patients is small. This should be stated clearly as a limitation of the study.

2. Is this the first investigating the distribution and expression of mast cells and/or chymase in IgG4-related disease and especially IgG4-ROD? Any relevant literature data should be mentioned briefly and discussed accordingly. For example, an increased number of IgE-positive mast cells has been observed in lymphoid, biliary, and pancreatic tissue samples of patients with IgG4-RD (1).

1. Culver EL, Sadler R, Bateman AC, Makuch M, Cargill T, Ferry B, Aalberse R, Barnes E, Rispens T. Clin Gastroenterol Hepatol. 2017 Sep;15(9):1444-1452.e6. doi: 10.1016/j.cgh.2017.02.007. Epub 2017 Feb 20.

Author Response

Feb 17, 2022

Editorial Office

Dear Dr. Reviewer 3:

Thank you very much for your E-mail of Feb 02 regarding our manuscript “Activation of Mast-Cell-Derived Chymase in the Lacrimal Glands of Patients with IgG4-Related Ophthalmic Disease” (Manuscript ID: ijms-1579696).

Thank you for your detailed review of our manuscript. We have revised our manuscript in accordance with your suggestions and recommendations, and have provided point by point responses to your comments below. We believe that the corrections will make our paper much more valuable and mature. Our responses to your comments are as follows:

1, Thank you very much for your valuable comments. As you pointed out, the sample in this study was too small. We have noted this in the Limitations paragraph (Lines :291-299).

Since surgical resection of LGs for the treatment of both severe canalicular obstruction and IgG4-ROD is rarely performed at our hospital, we could not enroll a satisfactory number of subjects in the present study. Additionally, the present study focused only on the features of the histological distribution and the degree of chymase expression between the two groups. However, since mast cells contain several mediators, possible differences in these other mediators between the groups might also have affected the study results. Therefore, one cannot reach a final conclusion until beneficial effects are identified using a chymase-specific inhibitor in animal models simulating the pathophysiology of IgG4-ROD.

2, Thank you very much for your valuable information related to the increase in mast cells and IgG4-related disease. So far, we have not yet found a report mentioning the relationship between mast cell-derived chymase and fibrotic LGS in patients with IgG4-related disease. However, since we feel the increase in mast cells in certain organs in IgG4-related disease would add valuable information to our present study, we have mentioned this in the Discussion paragraph and added the relevant literature to our references as follows (Lines 48-52):

Although the exact pathophysiology of IgG4-RD has yet to be elucidated, an autoimmune mechanism appears to be strongly related to this disorder [3]. For example, IgG4-RD is often associated with allergic conditions in many patients, and an increased number of IgE-positive mast cells has been observed in lymphoid, biliary and pancreatic tissue samples of patients with IgG4-RD (Culver et all, 2017).

Thank you very much for your kind consideration.

We look forward to the appearance of our revised manuscript in International Journal of Molecular Sciences.

Sincerely yours,

Yasushi Fujita, M.D.

Department of Ophthalmology, Osaka Medical and Pharmaceutical University, 2-7 Daigaku-machi, Takatsuki, Osaka 569-8686, Japan.

                     TEL: +81-72-683-1221 (Ext 8475)

                     FAX: +81-72-684-6730

                     E-mail: yasushi.fujita@ompu.ac.jp

Round 2

Reviewer 3 Report

No comments